# Hybrid Lime–Pozzolan Geopolymer Systems: Microstructural, Mechanical and Durability Studies

**DOI:** 10.3390/ma15082736

**Published:** 2022-04-08

**Authors:** Ariel Rey Villca, Lourdes Soriano, María Victoria Borrachero, Jordi Payá, José María Monzó, Mauro Mitsuuchi Tashima

**Affiliations:** Institute of Concrete Science and Technology (ICITECH), Universitat Politècnica de València, 46022 Valencia, Spain; arvilpo@posgrado.upv.es (A.R.V.); lousomar@upvnet.upv.es (L.S.); jjpaya@cst.upv.es (J.P.); jmmonzo@cst.upv.es (J.M.M.); maumitta@upvnet.upv.es (M.M.T.)

**Keywords:** hydrated lime, pozzolan, geopolymer, mortar, freeze–thaw cycles, water absorption

## Abstract

This work studies the possibility of using geopolymer materials to enhance the mechanical and durability properties of hydrated lime–pozzolan mixtures, which gave rise to the so-called “hybrid systems”. Two different waste types were used as pozzolan in the lime–pozzolan system: rice husk ash (RHA) and spent fluid catalytic cracking (FCC). The geopolymer fabricated with FCC was activated with commercial reagents (NaOH and Na_2_SiO_3_), and also with alternative sources of silica to obtain a lower carbon footprint in these mixtures. The alternative silica sources were RHA and residual diatomaceous earth (RDE) from the beer industry. The geopolymer mixture substituted the lime–pozzolan mixture for 30% replacement in weight. The hybrid systems showed better mechanical strengths for the short and medium curing ages in relation to the lime–pozzolan mixtures. Thermogravimetric analyses were performed to characterise the types of products formed in these mixtures. In the durability studies, hybrid systems better performed in freeze–thaw cycles and obtained lower capillarity water absorption values.

## 1. Introduction

Both the restoration and conservation of cultural heritage must be carried out to ensure compatibility with the original materials, while also meeting certain durability and mechanical strength standards.

The latter is important for those buildings constructed in seismic areas [1]. The heritage of cultures like the Roman one has reached our days, and many of these constructions were made with lime and natural pozzolan [2,3].

In recent decades, many studies have focused on restoring ancient buildings employing mixtures of lime with natural and artificial pozzolans [4,5,6,7]. Fernandez et al. [6] used metakaolin (MK) and natural volcanic material as pozzolans to prepare mortars with hydrated lime and hydraulic lime. They also added nano-TiO_2_ and perlite to the mixture. These authors concluded that adding nano-TiO_2_ and perlite produced lower compressive strengths than mortars with only lime and pozzolans but, in turn, the behaviour related to resistance to salts attacks was better.

Despite many publications having studied its properties, using lime–pozzolan mixtures is a less developed research line than the cement–pozzolan mixture. In the present work, two pozzolans for which studies endorse their use were tested: rice husk ash (RHA) and spent fluid catalytic cracking (FCC) residue.

RHA employed as pozzolan in mixtures with lime has been previously studied under different conditions. Pavia et al. [8] studied lime–RHA mixtures in proportions 1:0, 2:1, 1:1, 1:2 and 1:3 with water binder ratios of 0.86, 0.78, 0.70, 0.64 and 0.61, respectively, and at a constant sand/binder ratio of 1.5. They concluded that the 1:3 proportion conferred the mixture higher compressive strength at 28 curing days. This mortar reached 8.66 MPa compressive strength versus 0.86 MPa obtained by the 1:0 mortar for the same curing age. The presence of RHA accelerated setting time and enhanced bulk density. Méndez et al. [9] reported similar or slightly better results than Pavía et al. [8] using lime–RHA ratios of 1:1, 1:2 and 1:3 with constant water:binder and aggregate:binder ratios of 0.8 and 3. At 28 curing days, the authors obtained compressive strength close to 10 MPa for the 1:2 mortar. In all the systems, compressive strength increased with curing age, with 12, 18 and 18 MPa for systems 1:1, 1:2 and 1:3, respectively. Other authors have made mixtures with the RHA, MK and lime combination [10]. The presence of MK develops additional reaction products, such as hydrated gehlenite, which enhances the matrix and increases the absolute density of mixtures. The mortars with a 25% proportion of MK and 75% RHA and a 3:1 pozzolan:lime ratio obtained a compressive strength of 19 MPa at 28 curing days.

Arizzi and Cultrone [11] compared FCC and MK in lime–pozzolan mixtures. These authors substituted lime for MK and FCC at different percentages (10%, 15% and 20% per weight). The higher compressive strength of the mortars cured at 28 days was obtained for the mortar with 20% MK (9.12 MPa) versus the mortar with the same quantity of FCC, which only achieved 3.34 MPa. The authors established that the bigger FCC particle size was the reason for this pozzolan’s lower reactivity (the particle size of half the FCC volume went from 10 to 250 µm). García et al. [12] obtained better results by employing lime and FCC mixtures: the use of FCC was compared to RHA, but with a different lime:pozzolan proportion, with 1:1 for FCC and 1:3 for RHA. At 28 curing days, the mortar with FCC gave 10 MPa compressive strength and the RHA mortar achieved 8 MPa.

Other monuments that have remained in perfect condition throughout history, whose construction has aroused interest, are the pyramids of Egypt. Davidovits is the creator of the term “geopolymer”, and bases much of his research on the theory that the binder of these pyramids is mainly a mixture of natron, crushed limestone, clay, lime and water [13]. By applying this theory, Arcones et al. [14] conducted research using sodium carbonate, lime and MK as a binder material. The authors prepared two batches of mixtures in a constant proportion of 60% sand and 40% binder. In batch A, the water:binder ratio was 0.75 and the lime proportions were 5% and 10% substitutions of sodium carbonate and MK, respectively. In batch B, the water:binder ratio was 1.00 and the lime substitution percentages were 30%, 60% and 100%. The alkalinity of mixtures was limited and, consequently, the highest compressive strengths obtained for the mortars with 10% and 30% lime were around 8 MPa. Microstructural studies have demonstrated that the presence of lime in the binder promotes the formation of geopolymeric structures with the formation of N-A-S-H and N-(C)-A-S-H and the co-existence of C-S-H.

Using lime in alkaline activation mixtures has been explored in many research works. In the majority of them, the use of lime is limited to quantities between 1 and 20%, but normally at <10% [15,16,17,18,19]. Chakraborty et al. [15] used sewage sludge ash (SSA), quick lime and blast furnace slag (BFS) as precursors activated with NaOH. The best combination yielded a compressive strength of 31.3 MPa at 28 curing days with the mixture formed by 70% SSA, 20% quick lime and 10% BFS. Aziz et al. [16] activated natural pozzolan with lime (4–12% pozzolan replacement), and employed a mixture of NaOH and sodium silicate as an activator. Das et al. [17] studied the use of fly ash, silica fume and lime. All the papers generally demonstrated that limited quantities of lime enhance mechanical strength and shorten setting times.

This study proposes employing alternative geopolymeric materials (based on alternative silica sources such as RHA and RDE [20,21,22]) to enhance the early -mechanical strength of lime–pozzolan mixture throughout a new hybrid lime–pozzolan–geopolymer on the formation of hydrated-products in hybrid system. In the same way, durability aspects such as freeze–thaw cycles and capillarity water absorption were assessed.

## 2. Materials and Methods

Hydrated lime CL90-S (according to UNE-EN 459-1 [23]) was used to prepare mortars and pastes (supplied by Cales Pascual, Paterna, Spain). Siliceous sand with a fineness modulus of 4.1 was employed to produce mortars. NaOH pellets (98% purity, from Panreac S.L.U, Barcelona, Spain) and commercial Na_2_SiO_3_ (8% Na_2_O, 28% SiO_2_ and 64% H_2_O, from Merck S.L.U, Barcelona, Spain) were utilised to prepare alkaline-activating solutions.

This study employed: spent FCC residue, supplied by BP Oil España S.A.U (Grao de Castellón, Spain); RHA from Dacsa S.A (Tabernes Blanques, Spain); RDE, which came from Heineken España (Quart de Poblet, Spain). The chemical compositions of these materials are shown in Table 1.

These materials’ particle size distributions, as determined by a laser dispersion analysis, are depicted in Figure 1. FCC and RHA were previously milled in an industrial ball mill and respectively yielded a mean particle diameter of 17.12 μm and 20.30 μm. RDE was oven-dried at 105 °C for 24 h to remove moisture before using it. This material’s mean particle diameter was 46.40 μm.

Figure 2 shows some FESEM micrographs of the assessed materials. For FCC and RHA (Figure 2a,b, respectively), homogeneous powder material is observed due to the milling process. For RDE (Figure 2c), the typical skeletons of diatoms (microalgae) with a microporous structure are detected.

### Experimental Procedure

Table 2 summarises the mix proportions of the assessed mortars. All the mortars were produced at a set sand:binder ratio of 3. For traditional mortars (containing only lime and pozzolan), the binder is considered the sum of lime and pozzolan. For hybrid mortars, the binder is composed of 30 wt.% of FCC (precursor of the geopolymeric binder), and 70 wt.% of the binder is used for traditional mortars (lime–pozzolan mixture). This dosage was proposed in view of previous results obtained by the research group. In Table 2, the mix proportion of the assessed mortars is separately presented (lime–pozzolan binder and geopolymeric binder) to help to understand the mix proportions of hybrid systems.

The geopolymeric binder was produced using two different alkaline activating solutions: the first with commercial reagents (NaOH and Na_2_SiO_3_), and the second with alternative alkaline-activating solutions based on NaOH and alternative silica sources (RHA or RDE). All the activating solutions were prepared at a SiO_2_/Na_2_O molar ratio of 1.17, a water:binder ratio of 0.6 and 1.69 mol of Na_2_O per kg of precursor [24].

Otherwise, the lime–pozzolan proportions employed for the traditional mortars were set according to previously performed studies: a water:binder ratio of 0.8, and a lime/pozzolan proportion of 1:1 for lime:FCC and one of 1:2 for lime:RHA [9,12].

The nomenclature assigned to the traditional mixtures based on lime–pozzolan is “T-X”, where “T” indicates a traditional mixture and “X” is related to the used pozzolan: “FCC” or “RHA”. For hybrid systems, the adopted nomenclature is H-X/Y, where “H” indicates hybrid systems, “X” is related to the pozzolan used in the previously explained traditional mixture (lime–pozzolan) and “Y” denotes the type of alkaline-activating solution: “C“ stands for commercial solution (NaOH and Na_2_SiO_3_), “RHA” for an alternative solution using RHA as the alternative silica source and “RDE” for an alternative solution employing RDE as an alternative silica source. For example, a mortar named H-FCC/RHA represents a hybrid mortar using 70 wt.% of lime–FCC (from the traditional mortar) and 30 wt.% of FCC (a precursor of a geopolymeric binder) activated with an alternative alkaline-activating solution based on RHA.

Similarly, pastes with similar proportions (without sand), and cured under the same conditions, were prepared according to the microstructural analysis.

A universal test machine was used for the compressive strength tests of the cubic specimens (4 × 4 × 4 cm^3^) for 3, 7, 28 and 90 curing days. The TGA in pastes was carried out for 3, 28 and 90 curing days by TGA 850 Mettler-Toledo equipment. Experiments were performed at a heating rate of 10°C min^−1^ from 50 °C to 600 °C in a nitrogen atmosphere using pin-holed aluminium-sealed crucibles. The microstructure of the selected pastes was assessed by field emission electron microscopy (FESEM). Images were taken at 2kV and samples were previously carbon coated.

Prismatic specimens (4 × 4 × 16 cm^3^; cured for 90 days) were used for the durability tests. The capillarity water absorption coefficient (CWAC) was determined following the procedure described in UNE-EN 1015-18 [25]. The freeze–thaw test was run in a climate chamber following the recommendations adapted from UNE-EN 12,371 [26]. Thirty 6 h freeze–thaw cycles were performed, where freeze comprised 4 h (1 h at −10 °C and 3 h at −15 °C), and thaw comprised 2 h at 20 °C.

## 3. Results

### 3.1. Compressive Strength

The compressive strength development of all the assessed mixtures is depicted in Figure 3. The mechanical development of the T-FCC mortars is shown in Figure 3a. Compressive strengths of about 10 MPa and 15 MPa were achieved after 28 and 90 curing days, respectively. The T-RHA mortars underwent (Figure 3b) very slow mechanical development, and yielded about 12.5 MPa after 90 curing days. These results agree with the literature [12] and demonstrate the effectiveness of FCC compared to RHA mainly for the first 28 curing days.

The positive effect of hybrid systems versus traditional mixtures was evident. After 3 curing days, all the hybrid mixtures yielded similar or even higher compressive strength values than traditional mixtures (lime–pozzolan) after 90 curing days regardless of the employed pozzolan type.

Properties were enhanced for the hybrid binders containing lime–FCC. After comparing the effect of the alkaline-activating solution on the compressive strength of the H-FCC/Y systems, the RHA employed as an alternative silica source gave better results than RDE and the commercial reagents. Moreover, the hybrid mortars prepared using RDE as an alternative silica source yielded a similar compressive strength to the hybrid systems using commercial reagents for 3–28 curing days. These results are very interesting from an environmental point of view because of the high carbon footprint associated with Na_2_SiO_3_ reagent production [27]. The effect of the alternative silica source on geopolymeric systems has been reported in the literature. According to this study, RHA led to more enhanced compressive strength than RDE [21].

For the H-RHA/Y systems, a similar trend was observed independently of the type of alkaline-activating solution: commercial reagents, RHA and RDE as alternative silica sources. The maximum compressive strength (about 20 MPa) was achieved after 90 curing days for H-RHA/C. The fact that the results were similar was probably due to the smaller amount of amorphous Al_2_O_3_ employed in mixtures to form C-(N)-A-S-H gels.

The geopolymeric binder generally improved the compressive strength for the proposed hybrid systems, which obtained good values at 3–7 curing days. This fact could contribute to the large-scale application of these new hybrid binders to the conservation and restoration of cultural heritage.

In view of the results obtained, we can assume that another possible application of these mortars could be in the manufacture of simple prefabricated elements such as curbs, tiles, pavers, etc.

### 3.2. Durability Studies in Mortars

#### 3.2.1. Freeze-Thaw Cycles

All the specimens were water saturated for 24 h and then weighed under dry-surface-saturated (SSD) conditions before the freeze–thaw cycles. Specimens were measured after the 6th and 30th freeze–thaw cycles to make comparisons to the initial values (0 cycles). Figure 4 depicts mass variation (%) in relation to the initial mass of specimens after the 6th and 30th freeze–thaw cycles.

All the assessed systems presented incremented mass for the 6th freeze–thaw cycle vs. the initial mass to yield up to 12%. This was probably due to the entrained water in cracks/microcracks caused by water volume variation [28,29]. After the 30th freeze–thaw cycle, some specimens presented less mass variation than the samples after the 6th freeze–thaw cycle. This behaviour was evidenced by the reduced dimensions in the corners of some specimens (sample H-FCC/C), as Figure 5 illustrates Nevertheless, in most cases no significant mass variation occurred from the 6th to the 30th cycle, which indicates these specimens’ good stability.

Figure 6 represents the percentage of loss in compressive and flexural strengths. As we can see, flexural strength was affected more by freeze–thaw cycles than compressive strength. The percentages of loss mass in flexural strength for all the mortars were higher than 50%. Hybrid systems did not present enhanced behaviour compared to traditional systems.

The values of the flexural and compressive strengths of the mortars after the cycles are represented in Figure 7.

Compressive strength was less affected, and all the loss percentages were below 50%. Once again, hybrid systems achieved similar losses to traditional lime–pozzolan systems. No significant differences were found for using RHA or FCC. For comparison purposes, T-FCC yielded 10.22 MPa in compressive strength after 30 cycles, whereas H-FCC/RHA and H-FCC/RDE yielded 16.43 and 16.46 MPa, respectively. This finding means that the mechanical properties for the hybrid systems after the freeze–thaw cycles were significantly better than for the traditional hydrated lime–based system. Similarly to RHA, T-RHA had 6.92 MPa compressive strength after 30 cycles, whereas the corresponding hybrid systems with RHA and RDE obtained 12.04 MPa and 11.95 MPa.

The good behaviour found indicates that these mixtures would be suitable for applications in aggressive freeze–thaw environments.

#### 3.2.2. Capillary Water Absorption

Capillary water absorption is directly related to pore structure characteristics: size, pore shape, connectivity, among other factors [30]. An inverse relation between the capillary water absorption coefficient and compressive strength has been reported by several authors [31,32].

Figure 8 represents the capillary water absorption coefficient (CWAC) values of all the assessed systems.

As the results of the resistances revealed, the influence of using a setting accelerator had positive effects during the first 24 h of curing. At 28 curing days, both resistances came very close to the results of the mortars without the setting accelerator.

Throughout the entire series, the lime–pozzolan mixture values were higher than those of their respective hybrid mortars containing geopolymer. Lime–RHA was the mortar with the highest CWAC value. The CWAC values in the lime and lime–pozzolan mortars obtained by Veiga et al. [33] were 1.1–1.6 kg/(m^2^min^0.5^). For example, the value of the mortar with MK was 1.4 kg/(m^2^min^0.5^). These values are consistent with those obtained in the present work.

Figure 9 includes a comparison to corroborate this inverse relation between CWAC and compressive strength. Except for the CWAC value of 1.85 obtained by the lime–RHA mortar, we observe that the higher the compressive strength values are, the lower the obtained CWAC values.

### 3.3. Studies in Pastes

#### Thermogravimetric Analysis

Figure 10 presents the DTG curves for all the pastes at 3, 28 and 90 curing days. This figure also shows the lime–FCC paste in which four peaks are highlighted. Zone 1 (100–180 °C) is associated with loss of the combined water associated with calcium silicate hydrates (C-S-H). Zone 2 (180–240 °C) and Zone 3 (240–300 °C) are attributed to loss of combined water from calcium aluminate hydrates (C-A-H) and calcium aluminosilicate hydrates (C-A-S-H). Finally, Zone 4 (500–600 °C) is attributed to loss of water from the decomposition of hydrated lime [34]. The most developed area in the lime–FCC paste is associated with the formation of C-A-H and C-A-S-H because of the high pozzolan content. Unreacted hydrated lime was present in pastes until 28 curing days. At 90 days, the pozzolanic reaction had completely consumed all the lime. When geopolymer was added, the DTG of this hybrid paste differed from the lime–FCC paste. As depicted in Figure 10b–d, Zone 4 is attributed to the presence of lime, which disappeared after 3 curing days. The DTG curves of all the pastes fabricated with the commercial and alternative silica sources looked alike. In these pastes, the most developed peak changed to temperatures around 100–180 °C. Within this range, geopolymeric gels (N-A-S-H and N(C)-A-S-H) also lost combined water similarly to C-S-H. The N-A-S-H gel was produced by the reaction of Al_2_O_3_/SiO_2_ from FCC by alkaline activation [35]. In a paste in which a geopolymeric reaction takes place, the presence of calcium leads to N(C)-A-S-H formation [36].

It seemed evident that the geopolymeric reaction predominated when the geopolymeric binder was added to the binary lime–FCC system, and the presence of an alkaline activator promoted the formation of new products. These data agree with those obtained by other authors who confirmed in MK/lime pastes with a higher concentration of hydroxides (high alkalinity) that geopolymeric gel predominated over pozzolanic products, which became secondary products [37,38]. Regarding the conversion of N-A-S-H gels into C(N)-A-S-H, García-Lodeiro et al. [36] pointed out that, depending on the amount of calcium, total conversion into C-A-S-H gel could occur.

Figure 11a presents the DTG curves for the lime–RHA system. For these pastes, peaks only appeared in Zones 1 and 4, and the peak in Zone 4 was only visible at 3 curing days. This behaviour was because the quantity of lime was smaller than in that of the pastes with FCC. The principal product to form during the pozzolanic reaction was C-S-H [10]. In the hybrid pastes, the main peak was observed in Zone 1. Within this range, water loss from C-S-H, N-A-S-H and C(N)-A-S-H decomposition took place, and this technique did not allow them to be distinguished from one another.

## 4. Conclusions

The present research work demonstrates the viability of using various waste types from different sources as construction material. Hybrid mortars can be employed as restoration materials with better mechanical performance than pozzolan–lime mortars. We conclude that:-The hybrid binder improves compressive strength and yields good values at 3–7 curing days.-The systems with alternative silica sources obtain similar results to commercial reagents. This is a very important goal for achieving more environmentally friendly systems.-The CWAC values of the hybrid systems are lower than those of their respective traditional systems. This behaviour is positive in durability terms.-The mechanical properties after the freeze–thaw cycles for the hybrid systems are significantly better than for the traditional hydrated lime-based system.-The TGA performed with the cured pastes shows that the nature of cementing gels changes with the presence of geopolymeric binders, and the consumption of hydrated lime is completed at early curing ages because of C(N)-A-S-H gel formation.

## Figures and Tables

**Figure 1 materials-15-02736-f001:**
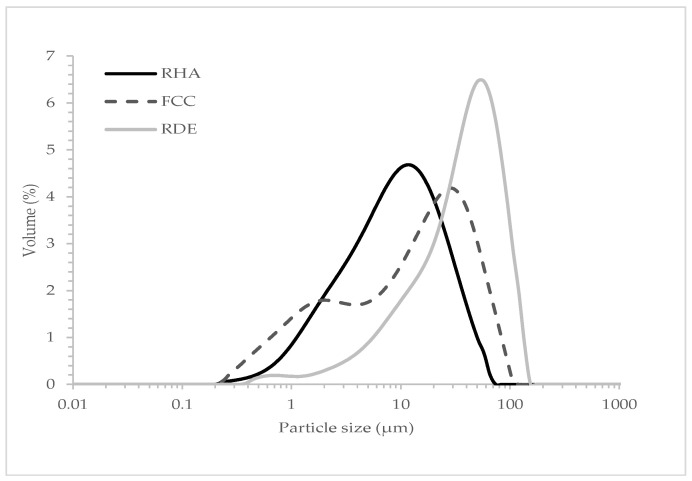
Particle size distribution curves of FCC, RHA and RDE.

**Figure 2 materials-15-02736-f002:**
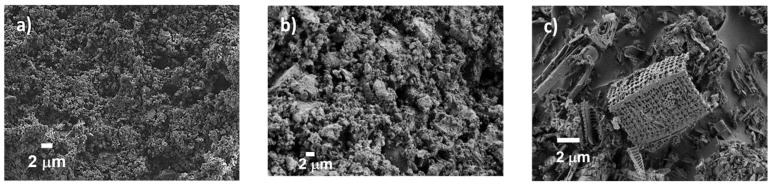
FESEM micrographs of materials: (**a**) FCC; (**b**) RHA; (**c**) RDE.

**Figure 3 materials-15-02736-f003:**
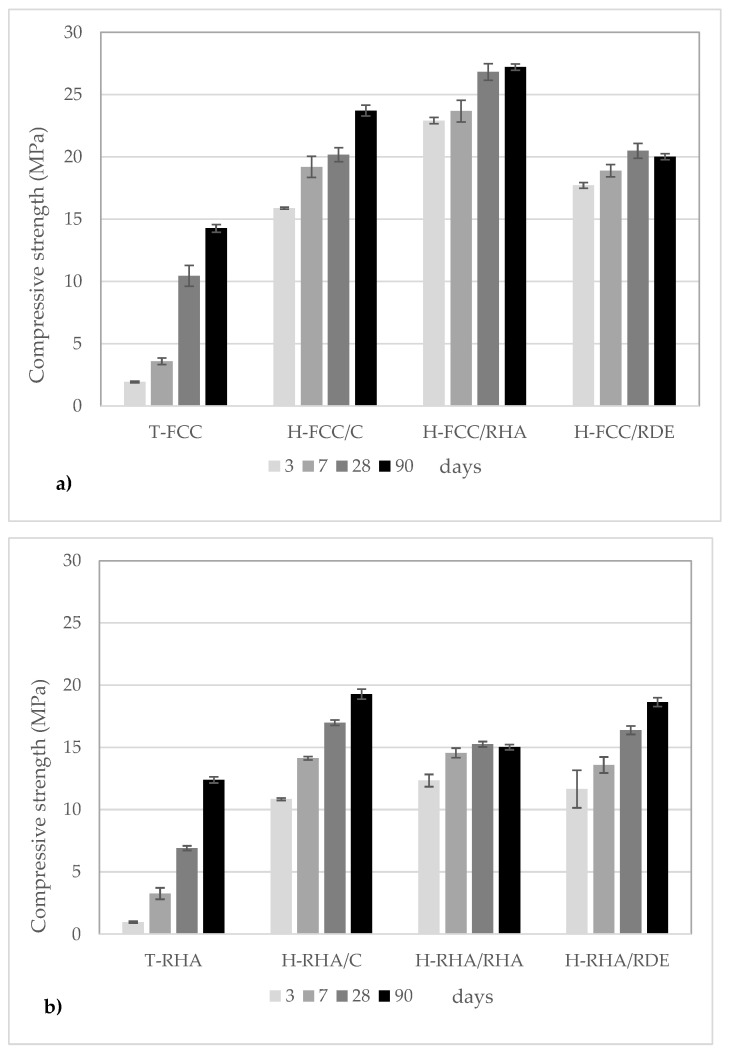
Compressive strength of mortars after 3, 7, 28 and 90 curing days: (**a**) FCC systems; (**b**) RHA systems.

**Figure 4 materials-15-02736-f004:**
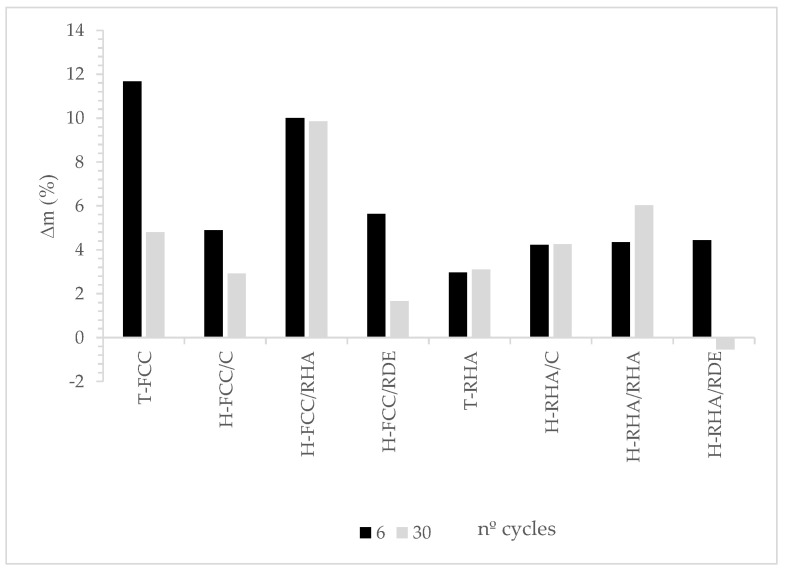
Mass variation (∆m, %) of all the assessed systems after the 6th and 30th freeze–thaw cycles.

**Figure 5 materials-15-02736-f005:**
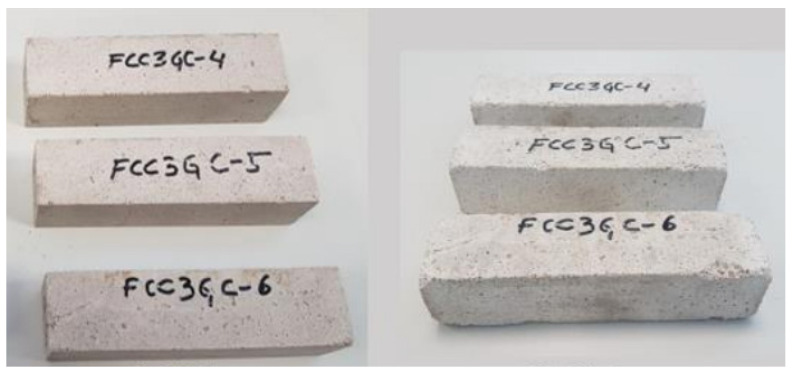
Photograph of the H-FCC/C mortar after 0 (**left**) and 30 (**right**) freeze–thaw cycles.

**Figure 6 materials-15-02736-f006:**
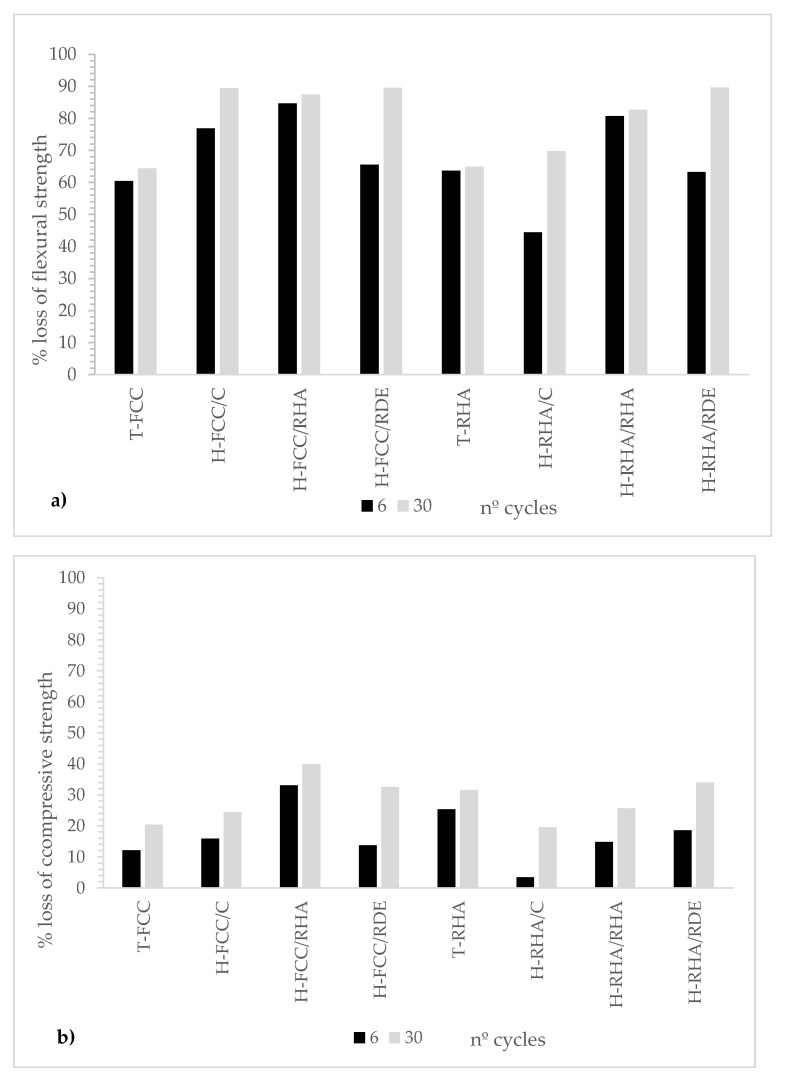
Mechanical properties after freeze–thaw cycles: (**a**) percentage of loss in flexural strength; (**b**) percentage of loss in compressive strength.

**Figure 7 materials-15-02736-f007:**
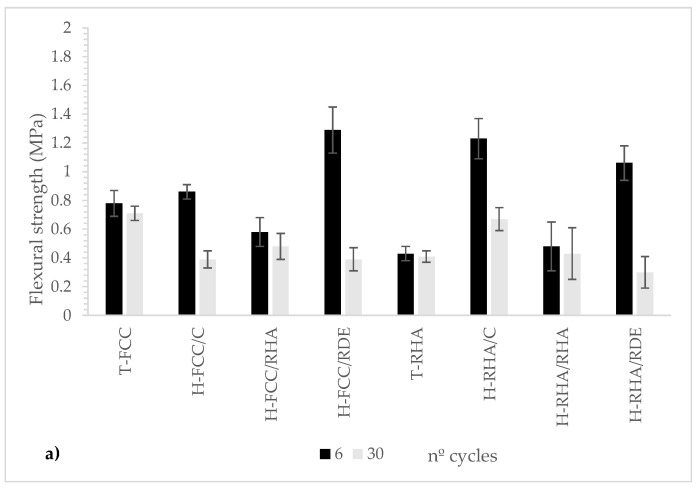
Mechanical properties after freeze–thaw cycles: (**a**) flexural strength; (**b**) compressive strength.

**Figure 8 materials-15-02736-f008:**
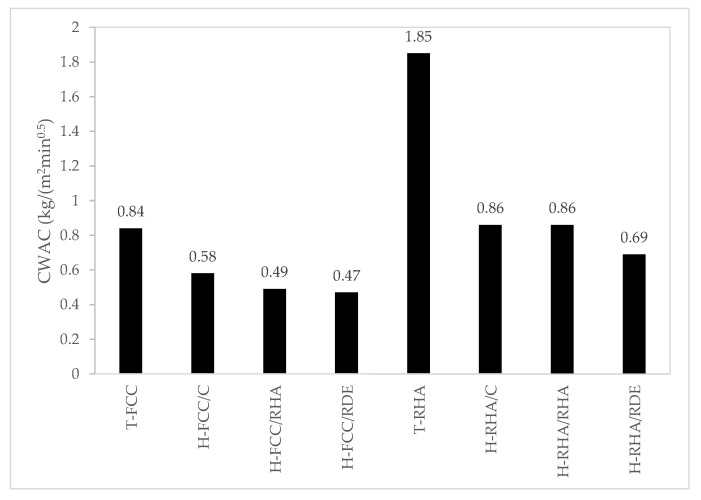
The capillary water absorption coefficient (CWAC) values (kg/(m^2^min^0.5^) of mortars.

**Figure 9 materials-15-02736-f009:**
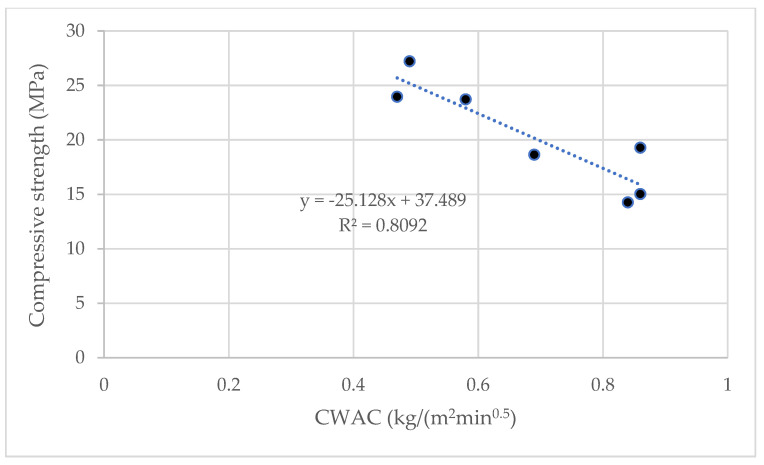
Relation between compressive strength (RC) and the capillary water absorption coefficient (CWAC).

**Figure 10 materials-15-02736-f010:**
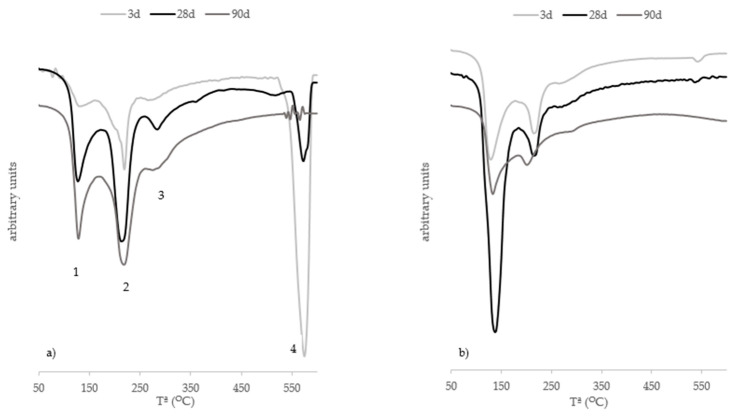
DTG curves of pastes at 3, 28 and 90 curing days: (**a**) T-FCC; (**b**) H-FCC/C; (**c**) H-FCC/RHA; (**d**) H-FCC/RDE.

**Figure 11 materials-15-02736-f011:**
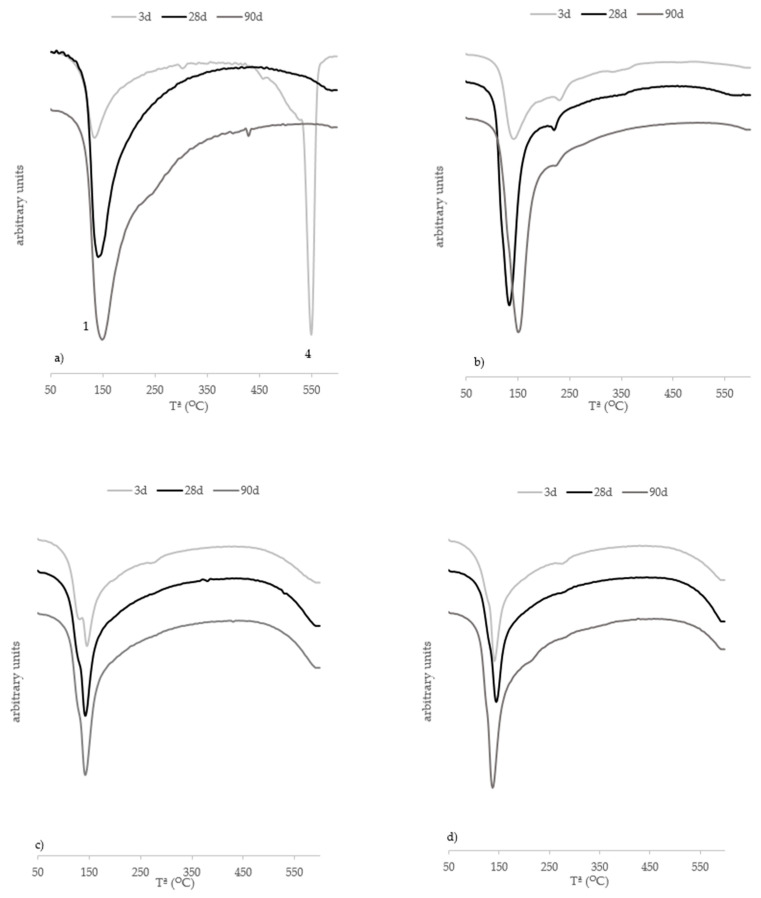
DTG curves of pastes at 3, 28 and 90 curing days: (**a**) T-RHA; (**b**) H-RHA/C; (**c**) H-RHA/RHA; (**d**) H-RHA/RDE.

**Table 1 materials-15-02736-t001:** Chemical compositions (wt%) of the spent FCC catalyst, RHA and RDE from the beer industry.

	Al_2_O_3_	SiO_2_	CaO	Fe_2_O_3_	K_2_O	Na_2_O	P_2_O_5_	MgO	SO_3_	Other	LOI
FCC	49.26	47.76	0.11	0.60	0.02	0.31	0.01	0.17	0.02	1.23	0.51
RHA	0.25	85.58	1.83	0.21	3.39	-	0.67	0.50	0.26	0.32	6.99
RDE	5.67	81.70	1.28	3.71	0.86	1.30	0.36	0.41	-	1.37	3.34

**Table 2 materials-15-02736-t002:** Mix proportions of traditional and hybrid mortars.

	Lime–Pozzolan Binder			Geopolymeric Binder					Sand
	Lime	Pozzolan	H_2_O	FCC	Alkaline-Activating Solution				
					H_2_O	NaOH	Na_2_SiO_3_	RHA or RDE	
T-FCC	262.5	262.5	420.0	-	-	-	-	-	1575.0
H-FCC/C	183.8	183.8	294.0	157.5	37.8	19.2	88.6	-	1575.0
H-FCC/RHA	183.8	183.8	294.0	157.5	94.5	37.8	-	27.6	1575.0
H-FCC/RDE	183.8	183.8	294.0	157.5	94.5	37.8	-	27.6	1575.0
T-RHA	175	350.0	420.0	-	-	-	-	-	1575.0
H-RHA/C	122.5	245.0	294.0	157.5	37.8	19.2	88.6	-	1575.0
H-RHA/RHA	122.5	245.0	294.0	157.5	94.5	37.8	-	27.6	1575.0
H-RHA/RDE	122.5	245.0	294.0	157.5	94.5	37.8	-	27.6	1575.0

## Data Availability

Data are contained within the article.

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
