# Peer review of "Hybrid Lime–Pozzolan Geopolymer Systems: Microstructural, Mechanical and Durability Studies"

_materials, 2022, doi:10.3390/ma15082736_

Round 1
Reviewer 1 Report
The authors propose geopolymeric materials to enhance the mechanical and durability properties of lime-pozzolan mixture. The authors shall address the following comments and revise their work accordingly.
- The title does not reflect the work well.
- The paper does not adequately justify why the purpose of the work is required.
- The language should be revised. There are several grammatical errors, as well as prepositions that are missing or incorrectly used.To illustrate, one preposition is missing and the other is incorrect in this sentence “The heritage of cultures like the Roman one has reached our days, and many of these constructions were made with lime and natural pozzolan.”. The correct version could be ”The correct use of prepositions in this sentence shall be “The heritage of cultures like the Roman one has reached to our days, and many of these constructions were made of lime and natural pozzolan.” I recommend that the authors seek the assistance of an English speaker or professional editor to revise the manuscript's language.
- Line 29 “This latter is important for those buildings…” correct as “The latter...”
- Remove the brand name of the equipment used from the manuscript.
- Discard 0 Cycles and 30 Cycles from Figure 5. It is sufficient to describe in the caption.
- The values displayed in Figure 6's chart are incorrect.
- The symbols used to represent degrees Celsius are incorrect.
- Discuss the work's theoretical and practical implications in greater depth.
Reviewer 2 Report
The authors' main objective is to study the effect on mechanical and durability properties of hydrated lime-pozzolan mixtures. However, the analysis of experimental results is not thorough enough.
Here are my comments:
- In the introduction, the demonstration of the significance of this study is not clear, the innovation is not prominent.
- The article does not explain why the proportion of using geopolymer mixture substituting the lime-pozzolan mixture is 30%.
- In the experiment, two kinds of materials with different particle sizes are used for comparative experiment, and whether the experimental difference caused by the particle sizes of materials as a variable are considered.
- The unit of particle size in line 129 of the article is "m". Please check whether there are any writing errors.
- RHA and FCC were used as pozzolan in the lime-pozzolan system for experimental research respectively. What is the purpose of the comparison? What are the guiding effects of the comparative experiment results for practical application?
Reviewer 3 Report
Review manuscript entitled: "Characterisation and durability studies on the new hybrid lime-pozzolan-geopolymer system"
The manuscript deals with the properties of hydrated lime-pozzolan mixtures that include geopolymer materials. The study investigates the feasibility to enhance the mechanical and durability properties of the hydrated lime-pozzolan mixtures. Mechanical properties such as compressive and flexural strength are investigated, at different ages as well as after 6 and 30 freeze-thaw cycles.
The main aim of the manuscript is to is to determine the mechanical properties of hybrid materials consisting of geopolymers and lime-pozzolan mixtures and to study the degradation mechanism of the materials when subjected to freeze-thaw cycles. The main contribution is the novel experimental data. These hybrid materials have not been tested previously as far as I know.
The paper uses a clear scientific approach to the subject matter, which is a clear strength. Furthermore, the text is well written and concise.
However, improvements are required in certain important aspects of the paper, along with some minor improvements:
- The title of the manuscript can be improved. The use of the term “on the new hybrid system” can be replaced by the most generic term “on new hybrid systems”, since the authors tested several mixtures and not just one.
- The authors should clearly mention why they chose this particular testing scheme. I can see the need for compressive and flexural tests, but why freeze thaw cycles? Do you anticipate a use for the material that is related to freeze thaw? The rational for this choice should be clearly mentioned in the end of the introduction.
- The authors should try to better describe what is shown in Fig. 4. Δm means increase of mass? If yes then Fig 5 show the opposite.
- The authors fail to report the data from the flexural tests. They only report in Fig. 6 the loss of flexural strength, but it is of interest to know the original values as well.
- I would suggest to add in Fig 7 the compressive strength as well, it would be easier to see the relationship between compressive strength and CWAC.
- It would be even more interesting to compare the porosity of the mixtures.
Round 2
Reviewer 1 Report
The authors responded to the majority of the comments. This comment is not addressed. "Discard 0 Cycles and 30 Cycles from Figure 5. It is sufficient to describe in the caption." or at least avoid the error marks under 0 cycle and 30 cycles (red wavy underlines).
Author Response
Review 1.
The authors responded to the majority of the comments. This comment is not addressed. "Discard 0 Cycles and 30 Cycles from Figure 5. It is sufficient to describe in the caption." or at least avoid the error marks under 0 cycle and 30 cycles (red wavy underlines).
The recommendation has been corrected in the figure 5 and have been discard the titles 0 and 30 cycles from the figure

Reviewer 3 Report
Review manuscript entitled: "Characterisation and durability studies on the new hybrid lime-pozzolan-geopolymer system"
The manuscript deals with the properties of hydrated lime-pozzolan mixtures that include geopolymer materials. The study investigates the feasibility to enhance the mechanical and durability properties of the hydrated lime-pozzolan mixtures.
Mechanical properties such as compressive and flexural strength are investigated, at different ages as well as after 6 and 30 freeze-thaw cycles.
The main aim of the manuscript is to determine the mechanical properties of hybrid materials consisting of geopolymers and lime-pozzolan mixtures and to study the degradation mechanism of the materials when subjected to freeze-thaw cycles. The main contribution is the novel experimental data. These hybrid materials have not been tested previously as far as I know.
The paper uses a clear scientific approach to the subject matter, which is a clear strength. Furthermore, the text is well-written and concise.
The authors have addressed almost all of my original comments.
I would request to change Fig. 6 back to the original Fig. I believe that the addition of the values on top of the bars complicates things.
It would be better if you could add a new Figure that will report the original flexural strengths (after 0 cycles), maybe something similar to Fig. 3.
Author Response
The authors have addressed almost all of my original comments.
I would request to change Fig. 6 back to the original Fig. I believe that the addition of the values on top of the bars complicates things.
It would be better if you could add a new Figure that will report the original flexural strengths (after 0 cycles), maybe something similar to Fig. 3.
The recommendation has been corrected and it has been included as figures 7a and 7b, graphs with the absolute values of the flexuras strength (7a) and compressive strength (7b), with their error bars, in a similar way to the form of figure 3, as the reviewer suggests.
When introducing a new figure, the following figures have been renumbered in the text and the figure caption
